Sobralia decora Bateman (Orchidaceae) and its relatives in South America

Baranow Przemyslaw przemyslaw.baranow@ug.edu.pl
Szlachetko Dariusz L.
Faculty of Biology, Department of Plant Taxonomy & Nature Conservation, University of Gdansk , Gdansk , Poland
Iriti Marcello
Electronic publication date: 2024 Sep 26
Publication date: 2024
Volume: 12
Electronic Location ID: e18078
Received 2024 Apr 10; Accepted 2024 Aug 20
Copyright: ©2024 Baranow and Szlachetko
Copyright year: 2024
Copyright holder: Baranow and Szlachetko
License: This is an open access article distributed under the terms of the Creative Commons Attribution License, which permits unrestricted use, distribution, reproduction and adaptation in any medium and for any purpose provided that it is properly attributed. For attribution, the original author(s), title, publication source (PeerJ) and either DOI or URL of the article must be cited.
License URL: https://creativecommons.org/licenses/by/4.0/

Keywords: Orchidaceae, Sobralia, Species identification, Taxonomy

Funding: The authors received no funding for this work.

==============================
A taxonomical study of Sobralia decora and allied species recorded in South America is presented. The group is characterized by light pink to purple flowers, often with a white tip of the labellum and yellow to brown area on its throat. If considering the habit, the species can be recognized and distinguished from all other Sobralia species by producing keikis–stems arising from the old inflorescence. The similarity of the species forming the discussed complex caused numerous mistakes noticed in the literature and among the herbarium specimens. The aim of the elaboration is to clarify the differences allowing to determine the living and herbarium specimens with no errors.

Introduction

The genus Sobralia Ruiz & Pav. was established in 1794 and since that time has become a large group of species consisting of over 200 taxa distributed in Central and South America.

The genus is known for its diverse morphology, particularly in inflorescence position and structure, as well as in floral bracts and flowers. This variability has been crucial for describing subgeneric taxa (Lindley, 1854; Reichenbach, 1873; Brieger, 1983).

Dressler (2002) extensively studied these groups and adopted Brieger’s classification. However, Dressler also noted that some Sobralia species possess unique morphological traits that do not fit into existing groups. The section that includes the type of Sobralia presented the most significant taxonomic challenge. According to Neubig et al. (2011) and Dressler et al. (2011), many species in this section formed a paraphyletic group more closely related to genera like Elleanthus C. Presl, Epilyna Schltr., and Sertifera Lindl. & Rchb. f. rather than other Sobralia sections. Moreover, members of the section including the generitype were morphologically distinct from other members of the genus.

These findings prompted taxonomists to reevaluate the genus and distinguish the section including the type of Sobralia as a separate genus. However, implementing these changes required several stages. The primary difficulty arose in handling the section, which retained the name Sobralia with S. dichotoma Ruiz & Pav. as the type species, while all other species needed reclassification under Cyathoglottis Poepp. & Endl., the oldest available synonym for Sobralia. This proposed reorganization posed practical challenges due to the large number of affected species, many of which are popular in horticulture as ornamental plants, potentially causing widespread confusion.

To address these concerns, Dressler et al. (2011) recommended conserving the name Sobralia with S. biflora Ruiz & Pav. as the conserved type species, aiming to uphold nomenclatural stability within the genus.

This allowed for the elevation of the previous section that includes the type of Sobralia to the rank of a separate genus—Brasolia (Rchb. f.) Baranow, Dudek & Szlachetko (2017). Since then, Sobralia has become more unified in terms of inflorescence structure. Its current section including the type species of the genus is characterized by a strongly shortened inflorescence with overlapping floral bracts. One of the groups distinguished within this section is a complex of species referred to as the Sobralia decora complex.

All species within this complex exhibit a unique vegetative feature that sets them apart from the rest of the genus—they produce keikis, which are shoots that emerge from old inflorescences. Often, multiple keikis can arise from a single inflorescence, creating the appearance of branching stems in these species. All the plants produce small or medium in size (as for the genus), white, pink or purple flowers, often with sepals lighter in color than the inner floral segments, with yellow to brown area on the labellum throat and with white tip of its apex. In the discussed group, there are no longitudinal keels running along the veins on the surface of the labellum, unlike in many other species of Sobralia. The similarity may be the reason of some misunderstandings of the species concepts and wrong identification of some of the herbarium collections. As an example we can give the plate no. 4570 published in Curtis (1851) described as S. sessilis Lindl. but presenting a specimen of S. decora Bateman with no doubts (Dressler, 2012).

The complex has no commonly used name. However, as the group have been studied in the area of Central America already by Dressler (2012) who named it as “Sobralia decora and its cousins”, we propose to give the unofficial name Sobralia decora complex to the species. As many herbarium specimens representing the complex that we had the opportunity to revise were named Sobralia decora (often incorrectly), we conclude that this name is the most appropriate for the group.

The unit doesn’t deserve the status of a separate taxon. Morphological features, especially the characteristic cone-like inflorescence, allow the group to be placed in the section including the type of the genus (former section Abbreviatae Brieger) without any doubts. The group is not well represented in phylogenetic analyses based on molecular studies (Neubig et al., 2011; Baranow, Dudek & Szlachetko, 2017; Baranow et al., 2022). These studies include only S. decora Bateman, S. sessilis Lindl., and S. yauaperyensis Barb. Rodr. In all published phylogenetic trees, these species are either close to each other in the same clade or form a completely separate group. It can therefore be inferred that, in addition to morphological similarities, they are also closely related. However, to confirm this, more species from the group should be studied and compared with the rest of the Sobralia species.

The presented results focus on the species within the complex found in South America, an area that has not been studied before. Although these species also occur in Central America, their variability in that region has already been analyzed and described (Dressler, 2012). This study is the first step toward a thorough examination of this species complex across its entire range. In this initial phase, we have combined our morphological studies results with existing literature. We hope that this will advance the understanding of the diversity of these species.

These species represent a genus that constitutes a significant component of the orchid flora in Central and South America, but they are also frequently cultivated by collectors and plant enthusiasts. We believe that understanding the variability within the complex will have practical implications that can be utilized both for understanding biological diversity and for individuals involved in the cultivation of these intriguing plants.

We present a list of species representing the group in South America. The complex is represented by six distinct species in the area –S. cataractorum Hoehne, S. decora, S. fenzliana Rchb. f., S. lowii Rolfe, S. sessilis and S. yauaperiensis. We clarify their diagnostic features to facilitate the determination of their specimens. The most significant taxonomic characters are used in the determination key and are presented in the detailed morphological description of each species. To facilitate the comparison of the studied taxa, we present the most remarkable differences in a table (Table 1).

Material and Methods

The presented study is based on the examination of herbarium collections representing the discussed taxa deposited at AMES, B, COL, COAH, CR, F, K, MA, MO, P, RPSC, U, US and W (herbarium acronyms according to Thiers, 2024). The manuscript includes citations of 64 collections listed in Material S1. Morphological analyses were proceeded with the examination of the diagnoses and original illustrations. The distribution, altitude, ecology and habitat of the studied taxa are presented on the basis of data taken from the herbarium labels which are also available as Material S2.

Morphological studies based on herbarium material begin with photographing the plant, and measuring its stems, leaves, and floral bracts. The surfaces of the leaf sheaths and leaf blades are also observed. This study includes only those collections among the examined samples that contain flowers suitable for morphological analysis. Flowers are rehydrated by boiling in water, after which their elements are measured. The labellum is observed for the presence of thickenings or other potentially significant features. All flower parts are documented in a working drawing that aims to represent their dimensions and shape.

Taxonomic treatment

Key to the species

1. Column apical stelidia long and distinctly exceeding the column apex …S. fenzliana	
1* The apical stelidia of the column either slightly exceeding or being below the column apex …2	
2. Labellum base with marsupiform thickening …S. cataractarum	
2* Labellum base with a pair of fleshy or keel-like calli …3	
3. Plant over 100 cm tall …4	
3* Plant up to 60 cm tall …5	
4. Leaf sheaths dark-hirsute …S. sessilis	
4* Whole stems smooth …S. yauaperyensis	
5. Leaf sheaths covered by black, minute hairs, sepals ca. 50 mm long …S. lowii	
5* Leaf sheaths green, nearly glabrous, sepals up to 40 mm long …S. decora	

Sobralia fenzliana Rchb.f. (Fig. 1A)

Bot. Zeitung (Berlin) 10: 714. 1852; Type: Panama. Warszewicz J. 48 (Holotype: W!; AMES! - illustration of type).

Plant 100–200 cm tall. Stem caespitose, erect or ascending, sometimes pendent, often forming keikis on old inflorescences, enclosed by furfuraceous leaf sheaths. Leaves 5–20 cm long, 2−6.5 cm wide, elliptic to broadly elliptic-lanceolate, acute to abruptly acuminate, glabrous except the base of dorsal surface, rather thin in texture. Floral bracts imbricating, setose or lepidote. Flowers magenta with whitish tips of floral segments, labellum throat reddish. Dorsal sepal up to 60 mm long, 15 mm wide, ligulate to oblong-oblanceolate, acute. Lateral sepals up to 60 mm long and 15 mm wide, ligulate to oblong-lanceolate, acute, oblique. Petals 60 mm long, 20 mm wide, narrowly elliptic-obovate, acute, margins somewhat undulate. Labellum 60 mm long, 40 mm wide, obovate above cuneate base, obtuse, shallowly bilobed at apex, without any calli, margins entire in basal half, irregularly undulate. Column up to 30 mm long, apical stelidia erect, subacute, distinctly exceeding the gynostemium apex.

Figure 1 Flowers morphology of the southamerican members of Sobralia decora complex.

(A) Sobralia fenzliana Rchb. f. a–dorsal sepal, b–lateral sepal, c–petal, d–labellum. Drawn from Sine coll. (W-R). (B) Sobralia cataractorum Hoehne. a–dorsal sepal, b–lateral sepal, c–petal, d–lip. Redrawn from (Hoehne, 1945). (C) Sobralia sessilis Lindl. a–dorsal sepal, b–lateral sepal, c–petal, d–labellum. Drawn from Maguire & al. 53906 (U). (D) Sobralia yauaperyensis Barb. Rodr. a–dorsal sepal, b–lateral sepal, c–labellum. Redrawn from (Hoehne, 1945). (E) Sobralia lowii Rolfe. a–dorsal sepal, b–lateral sepal, c–petal, d–labellum. Redrawn from the illustration kept at K (11812_75). (F) Sobralia decora Bateman. a–dorsal sepal, b–lateral sepal, c–petal, d–labellum. Drawn from Szlachetko s.n. (UGDA). Scale bar–10 mm.

Habitat and ecology. Epiphytic or terrestrial in rain forest, low tree forest at the edge of savanna, margin of remnants of forest.

Distribution: Nicaragua, Costa Rica, Panama, Colombia, Ecuador. Alt. 300–2000 m.

Representative specimens: COLOMBIA. Caquetá. Sierra de Chiribiquete. Bosque cerca de la cueva de las pinturas (levantamiento 21), 30 Nov 1992, Palacios P. 2880 & Barbosa C., Cortes R., Rangel O. (COL!); Sierra de Chiribiquete, 1°4′19″N, 72°40′5″W, En escarpes altos, Alt. 810 m. 29 Nov 1992, Barbosa C. 8109 & Cortes R., Palacios P., Rangel O. (COL!). Vaupés. Alto Cuduyari, savanna Yapobodá, Dec 1943, Allen P.H. 3156 (COL!); Mitú and vicinity, lower río Kubiyu, 25 Sep 1976, Zarucchi J.L. 2140 (COL!; K!, MO!). ECUADOR. Cotopaxi. La Mana Canton, between Guayacan (13.1 km N of “Lan Mana”) and Montenuevo (N of Pacayacu), at end of road which branches to the right 23.6 km from Guayacan, in vicinity of Escuela Quindigua, 10.7 km beyond the junction in road to Escuela Quindigua, 0°39′S, 79°05′W, Alt. 1480-1530 m. 9 Apr 1992, Croat T.B. 73788 (MO!); Manabi. Montecristi, Cerro Montecristi. Carretera Manta-Jipijapa, entrada por Montecristo o El Chorrillo, 1°02′S, 80°41′W, Alt. 300–600 m. 11-12 Nov 1995, Nunez T., Sandoval S. & Machuca J. 379 (MO!); Manta, Bosque Protector Chorillos, 1°17′18”S, 80°37′05”W, Alt. 590 m. 14 Dec 1999, Linder S. & Grupo Post-Grado MO-QCNE 487 (MO!); Pichincha. Km 68, Quito-Santo Domingo via nueva carretera por Tandapi, Alt. 2000m. 8 Mar 1985, Dodson C.H. & P.M. 15665 (K!); Quito. Cerro Antisana, Shinguipino, between Rios Napo and Tena, 8 km SE of Tena, 0°30′S, 78′W, Alt. 400m. 29 Aug 1960, Grubb P.J., Lloyd J.R., Pennington T.D. & Whitmore T.C. 1501 (AMES!, K!); Zamora-Chinchipe. Along road between Zumbi on Rio Zamora and summit of Cordillera del Condor beyond Paquisha, 10.1 km beyond Ro Nangaritza Bridge, 29.1 km E of Zumbi, 3° 56′13”S, 78°37′27″W, Alt. 1352 m. 16 Jul 2004, Croat T.B., Hannon L.P., Walhert G.A. & Katan T. 91129 (MO!).

Table 1 Comparison of the most significant taxonomic characters of the studied species.

	Sobralia fenzliana	Sobralia cataractorum	Sobralia yauaperyensis	Sobralia sessilis	Sobralia lowii	Sobralia decora	
Stem size (cm)	100–200	100–00	90–120	Over 100	30–45	30–60	
Leaves and leaf sheaths surface cover and colour	Leaves glabrous except the base of dorsal surface, leaf sheaths furfuraceous	Smooth	Smooth	Thickly covered with dark hairs as to have quite a purplish cast	Leaves lower side and leaf sheaths minutely hirsute	Nearly smooth	
Flowers colour	Magenta with whitish tips of floral segments	Intensive lilac red, little paler outside, lip with yellow throat	Flowers white, deep purple or rosy, lip darker than other floral segments, with basal part yellowish	White to deep purple, lip darker than other segments, with yellow throat	Bright purple	Sepals and petals white or pale pink, lip pink inside with pale reddish brown throat and white spot at the apex	
Flower segments length (mm)	Up to 60	Lip up to 65, other segments up to 50	Petals and lip—44–50, sepals—44–54	44–54	Labellum and petals—45, sepals—50	35–40	
Lip basal calli	No	Marsupiform thickening	A pair of short, lamellate calli	A pair of lamellate calli	Fleshy, linear calli	A pair of fleshy, ridge-like calli	
Column size (mm)	Up to 30	ca 35	25–30	25–30	ca 40	ca 15	
Apical stelidia	Distinct, long	Short, strongly falcate	Short, recurved	Short, strongly falcate	Short, strongly falcate	Short, strongly falcate	

Notes. Sobralia fenzliana can be distinguished from other members of the Sobralia decora group by the apical stelidia of the gynostemium, which are distinctly longer and exceed the apex of the column more markedly compared to other species. In contrast, all other species within the group possess short, strongly falcate stelidia that either do not exceed or only indistinctly exceed the column apex (Dressler, 2012). Another distinctive feature of S. fenzliana is the absence of a pair of calli at the base of the labellum. While Garay (1978) illustrates a pair of labellum basal thickenings in the Flora of Ecuador, analysis of the type material deposited at W and details of the type specimen drawing at the AMES herbarium reveal that this species lacks such basal calli. This feature clearly necessitates further analysis based on a larger sample size, including the living specimens.

Sobralia cataractarum Hoehne (Fig. 1B)

Comm. Lin. Telegr., Bot. 1: 39. 1910; Type (Engels, Silva & Koch, 2021): BRAZIL. MATO GROSSO. without exact municipality, surrounding the Salto Sepotuba [Commisão das Linhas Telegráficas Estratégicas do Matto Grosso ao Amazonas]. Mar 1909 [fl.]. Hoehne F.C. 1686 (Lectotype: R 2988); ibidem, April 1909, [fl.], Hoehne F.C. 1823 (Syntype: R - lost).

Slender, erect plant 100–200 cm tall, stem robust, producing keikis, smooth, in upper part alternately, bilaterally leafy. Leaves 17–22 cm long, 4–7 cm wide, stiff, very flexuous, broadly ovate, acute or acuminate, base shortly pointed or sub-rounded to the apex of the green sheath, 7, sometimes 5-nerved. Bracts opposite, overlapping, quite unequal in length, narrowly triangular, folded, foliaceous. Flowers intense lilac red, a little paler on the outside, labellum with yellow throat, segments slightly spreading at the apex. Sepals basally connate into a long tube. Dorsal sepal 50 mm long, 25 mm wide, lanceolate, acute. Lateral sepals 50 mm long, 25 mm wide, lanceolate, acute, oblique. Petals 50 mm long, 25 mm wide, subspathulate oblong, a little widened at the top and subabruptly attenuated, very shortly acuminate, the margin quite wavy. Labellum 65 mm long, 35 mm wide, obovate-oblong in outline when spread, saccate in the middle, enveloping the column basally, with a marsupiform thickening at the base, the margins folded-waved on the top. Column 35 mm long, erect or slightly curved. Apical stelidia acute, curved.

Habitat and ecology. Terrestrial or litophytic along river banks. Flowering March –April.

Distribution. Brazil.

Notes. The species remains poorly known, currently represented by a single identified collection. Engels, Silva & Koch (2021) conducted a detailed study based on this collection and classified it as a synonym of Sobralia sessilis. We acknowledge that the examined herbarium material is indeed very similar to S. sessilis. However, upon reviewing both the literature and the protologue of S. cataractorum, we must disagree with the conclusion drawn by Engels, Silva & Koch (2021). They noted that S. cataractorum shares the type of surface of the leaf sheaths with S. sessilis. Nevertheless, literature data and the protologue indicate that S. sessilis has dense black hairs on the leaf sheaths, whereas S. cataractorum was described as having green, smooth leaf sheaths.

Moreover, a significant difference is observed in the basal protuberances of the labellum between the two taxa. Engels, Silva & Koch (2021) claimed that the protologue of S. cataractorum does not provide information about the labellum base structure. However, Hoehne (1945), the author of the protologue, published additional descriptions describing a basal thickening on the labellum of S. cataractorum. Engels, Silva & Koch (2021) suggested that this thickening is similar to the protuberances observed in S. sessilis. A comparison of Hoehne’s description with his characterization of S. sessilis in the same work reveals a distinct difference: S. cataractorum has a single marsupiform basal thickening, while S. sessilis features two distinctly separate, elevated lamellae.

Given these differences, we propose to maintain S. cataractorum as a separate species and not include it in the synonymy of S. sessilis.

Sobralia sessilis Lindl. (Figs. 1C, 2, 3, 4).

Edwards’s Bot. Reg. 27: Misc. 3. 1841; Type: Venezuela. Schomburgk R. s.n. (Holotype: K-L!). ≡ Cattleya sessilis (Lindl.) Beer, Prakt. Stud. Orchid.: 214. 1854.

= Sobralia panamensis Schltr., Repert. Spec. Nov. Regni Veg., Beih. 17: 11. 1922; Type (Christenson 1991: 131): Panama. Powell C.W. 21 (Lectotype: AMES! 00090618; Isolectotypes: AMES! 00090617, K!, MO! 955891, MO! 961330).

=Sobralia panamensis Schltr. var. albiflos Schltr., Repert. Spec. Nov. Regni Veg., Beih. 17: 11. 1922; Type (Christenson 1991: 131): Panama. Powell C.W. 31 (Lectotype: AMES! 00090619; Isolectotype: K!).

Stem over 100 cm tall, erect, leafy mainly in the upper part, often with keikis growing from the old inflorescences, covered by dark-hirsute sheaths. Leaves up to 22 cm long and eight cm wide, oblong- or elliptic-lanceolate, long-acuminate. Flowers white or deep purple or rosy, labellum darker than other floral segments, with basal part yellowish. Dorsal sepal 44–54 mm long and 13–15 mm wide, elliptic-lanceolate, acute. Lateral sepals ca 44–54 mm long, 15–18 mm wide, oblong lanceolate, acute. Petals ca 44–50 mm long, 14–17 mm wide, elliptic-lanceolate, widely acute to rounded. Labellum 44–50 mm long, 28–32 mm wide, broadly elliptic to obovate or obovate-rhomboid, slightly retuse and undulate at apex, with a pair of short, white, lamellate calli at the base, neither thickenings along the veins, nor any other protuberances on the center of lip. Column 25–30 mm long, apical stelidia short, strongly recurved. The middle of the column with two raised keels in front.

Figure 2 Habit of Sobralia sessilis Lindl., drawn from Maguire & al. 53906 (U). Scale bar –50 mm.

Figure 3 Habit of Sobralia sessilis Lindl. (Mauro Rosim, Brazil).

Figure 4 Flower of Sobralia sessilis Lindl. (Mauro Rosim, Brazil).

Habitat and ecology: Epiphyte, growing usually along rivers and creeks, terrestrial on savannas. Flowering throughout the year.

Distribution: Colombia, Venezuela, Guyana, Suriname, French Guiana, Ecuador, Peru, Brazil, Bolivia. Alt. from sea level up to 1400 m.

Representative specimens: COLOMBIA. Amazonas. basin of Rio Negro, Rio Tikkie, 5 May 1942, Froes R. 12555/24 (AMES!); Amazonas/Vaupes. Rio Apaporis, Sorotama (above mouth of Rio Kananari) and vicinity, 0°5′N, 70°40′W, Alt. 900ft. Jan 1952, Schultes R.E. & Cabrera I. 19841 (AMES!). Guaviare. Mpio. San José del Guaviare. Serrania de la Lindosa, sector entre La Lindosa-Charco Indio-Pozos Naturales-La Rebecca, 2°30′28″N 72°38′31″W, 5 Aug 2007, Cardenas D., Chols O. & Lopez A. 29857 (COAH!); Mpio. San José del Guaviare. Serrania de la Lindosa. Sector los Alpes, 2°29′22.8″N 72°46′54.1″W, Alt. 300–400 m. 9 Sep 2008, Cardenas D., Castano N., Zuluaga A. & Sedano J. 21801 (COAH!); Mpio. San José del Guaviare. Serrania de la Lindosa. Cerros de la Fortaleza y Monserrate, sector SE de la serrania, 2°27′49.3″N 72°37′8.3″W. Alt. 300–400 m. 6 Sep 2008, Cardenas D., Castano N., Zuluaga A., Rodriguez H. & Lucena A. 21727 (COAH!). Meta. Sabanas de San Juan de Arama, margen izquierda del río Guejar, alrededores del aterrizaje “Los Micos”. Alt. 500 m. 26 Jan 1951. Idrobo J.M. & Schultes R.E. 1326 (AMES!, COL!, US!). Vaupés. Río Kuduyari (affluent of río Vaupés), Yapapodá, 4-6 Oct 1951, Schultes R.E. & Cabrera I. 14321 (MO!); Mpio La Macarena. Vereda El Billar, afloramiento rocoso anexo al Cerro La Antena, margen izquierda del río Guayabero, aguas abajo, 2°13′26.6″N 73°48′31.2″W, Alt. 280–350 m. 4 Nov 2002, Rivera D., Cardenas D., Lopez R., Ramirez J.G. & Polanco P. 1214 (COAH!). VENEZUELA. Bolivar. Sierra de Lema, Cabeceras of Rio Chicanan, 80 km en linea recta al. Suroeste de El Dorado, 6°5′N, 62°W, Alt. 700 m. 22 Aug 1961, Steyermark J.A. 89426 (AMES!); Sierra Ichun: ladereas boscosas al. Norte del Salto Maria Espuma (Salto Ichun), a lo largo del rio Ichun (tributario del Rio Paragua), 4°46′N, 63°18′W, Alt. 625–725 m. 27 Dec 1961, Steyermark J.A. 90280 (US!). Delta Amacuro. Inundated forest Rio Cuyubini, along lower section of river, upstream from Casa Cuyubini, Alt. 90 m. 12 Nov 1960, Steyermark J.A. 87479 (AMES!). Zulia. Sierra de Perija, Bosque entre la loma que conduce hacia el Pishikako y Pishikako siguiendo el Rio Vikay-kuna (afluente del Rio Tumuriasa), cerca de la frontera Colombo-Venezolana, Alt. 1400 m. 31 Mar 1972, Steyermark J.A. & Dunsterville G.C.K. 105763 (AMES!). Flowered by Messrs Loddiges, received from Schomburgk R. s.n. (K-L!). GUYANA. Barima-Waini Region. Upper Sebai River, tributory of Kaituma R. 8 km upriver from Sebai river, 7°51′N, 59°17′W, Alt. 10 m. Hoffman B., Capellaro C. & Benjamin T. 671 (US!). Sipaliwini. Tafelberg, savannas no. 2 and 3, 9 Aug 1944, Maguire 24262 (K!); Zuid River, vicinity Falls Camp, 2 km above confluence with Lucie River, 220 m a. s. l., 3°20′N 56°49′W−3°10′N 56°29′W, 1 Jul 1963, Maguire et al. 53906 (U!); Lucie River, ca 2 km below affluence of Oost River, 225 m a. s. l., 3°20′N, 56°49′W– 3°32′N, 56°26′W, Maguire et al. 54143 (NY!). Upper Essequibo. Rewa River, near Corona Falls, 2°10′N, 58°40′W, Alt. 160 m. 5 Sep 1999, Jansen-Jacobs M.J., ter Welle B.J.H., Haripersaud P.P., Muller O. & van der Zee E. 2000 (U!); near Corona falls, 3°10′N, 58°40′W, Alt. 160 m. 5 Sep 1999, Jansen-Jacobs M.J., ter Welle B.J.H., Haripersaud P.P., Muller O. & van der Zee E. 5791 (B!, MO!, P!, US!). SURINAME. Morowijne. Coermotibo, 2 km W od mouth of Wanekreek, 30 Jan 1979, Teunissen M.P. 1064 (U!). Para. Parariver near Wlegelegen, 6 Jun 1971, Westra L.Y.T. 1018 (U!); Forest bank of Zuidriver near Lucie R. 2 Jul 1963, Schultz J.P. 10017 (K, U!). FRENCH GUIANA. Monts D’Arawa, Pied du versant sud-est de la grande savane-roche. Rive droite de la crique, 2°49′00″N, 53°22′00″W, Alt. 200 m. 10 Jul 2002, de Granville J.J., Crozier F. & Sarthou C. 15193 (P!). ECUADOR. El Oro. Vicinity of Portovelo, 6-15 Oct 1918, Rose J.N. & Rose G. 23947 (US!). PERU. Loreto. Rio Fuvineto, affluent du Rio Putumayo. Rerritoire des indiens Secoya, 27 Dec 1977, Barrier S. 213 (P!); Rio Yuviento, affluent du Rio Putumayo. Terr. Des indiens Secoya. Village de San Martin sur le Yubineto en amont de Bellavista, 31 Dec 1977, Barrier S. 221 (P!). BOLIVIA. La Paz. Coroico, Rio Yolosa, Alt. 1400 m. 23 Sep 1963, de la Sota E. 20467 (K!). BRAZIL. Bahia. Mun. Morro do Chapeu, Proximo ao leito do rio Ferro-doido, na mata, 11°37′41″S, 41°00′04″W, Alt. 1000 m, 13 Mar 1996, Conceicao A.A. et al. 2336 (K!); 19.5 km SE of the town of Morro do Chapeu on the BAO52 road to Mundo Novo, by the Rio Ferro Doido, 41°02′W, 11°38′S, Alt. 900 m. 2 Mar 1977, Harley et al. 19238 & 19239 (K!); Roraima. Vicinity of Auaris, 4°3′N, 64°22′W, 10 Feb 1969, Prance G.T. & al. 9803 (K!).

Notes. Dressler (2012) has noted that Sobralia sessilis is very similar to S. fenzliana. However, the two species can be distinguished by the shape of the apical stelidia of the gynostemium. In Sobralia sessilis they are distinct, but short, strongly falcate, slightly extending beyond the gynostemium apex. The stelidia of S. fenzliana are long, erect and distinctly extend the apex of gynostemium.

Sobralia panamensis, indicated here as a synonym of S. sessilis, merits attention due to its problematic status when considering the orchid floras of Central and South America. In some publications, this taxon has been listed as a synonym of S. decora (e.g., Kolanowska, 2014; Pupulin, 2002; Szlachetko et al., 2020), while other sources present it as a synonym of S. fenzliana (e.g., Bogarín et al., 2014; Garay, 1978).Our study of the protologue and diagnosis resulted in a conclusion that S. panamensis should be treated as a synonym indeed, but it is a synonym of S. sessilis with no doubts. The protologue suggests that Sobralia panamensis was described based on a comparison with S. sessilis. The only differences between the two taxa indicated by the author are a somewhat different shape of the labellum and its throat, which is narrower in S. panamensis. After examining the accessible collections of the entire complex, we conclude that its species demonstrate some variation, particularly in the shape of the floral segments, especially the lip. (which is also difficult in preparation as it is concave in the central part and it is a challenge to measure it and define its shape precisely) and we do not see the reason to distinguish it as a separate species. At the same time, some other features allow to exclude the possibility of treating S. panamensis as S. decora. Sobralia panamensis was described as much taller plant (90–110 cm vs 30–60 cm), with larger floral segments (50–60 mm vs ca. 35-40 mm) and leaf sheaths surface similar to the one observed in S. sessilis vs nearly smooth leaf sheaths in S. decora). It cannot be also recognized as conspecific with S. fenzliana which is easy to recognize thanks to the characteristic long apical stelidia of gynostemium. Examination of the flowers of S. panamensis type collection has shown that the structure of the stelidia is clearly different than in S. fenzliana.

Sobralia yauaperyensis Barb. Rodr. (Fig. 1D)

Vellozia, ed. 2, 1: 131. 1891. TYPE: Brazil, Barbosa Rodrigues s.n. (not localized)

Stem 90–120 cm tall, glabrous, flexuose, leafy mainly in the upper part, often with keikis growing from the old inflorescences. Leaves up to 16 cm long, five cm wide, oblong-lanceolate, acuminate, membranaceous to coriaceous. Inflorescence terminal, cone-like, with one flower at a time. Floral bracts 2–6 cm long, lanceolate, concave, acute. Flowers white, deep purple or rosy, labellum darker than other floral segments, with basal part yellowish. Sepals connate basally. Dorsal sepal 44–54 mm long, 13–15 mm wide, lanceolate, acute. Lateral sepals ca. 44–54 mm long, 15–18 mm wide, lanceolate, acute. Petals ca. 44–50 mm long, 14–17 mm wide, broadly lanceolate, widely acute to rounded. Labellum 44–50 mm long, 28–32 mm wide, oblong, with a pair of short, white, lamellate calli at the base, neither thickenings along the nerves, nor any other proturberances on the centre, margins somewhat crispate, apex retuse, recurved. Gynostemium 25–30 mm long, apical stelidia short, recurved, acute.

Ecology: Epiphytic or terrestrial.

Distribution: Guyana, French Guiana, Brazil.

Representative specimens:

GUYANA. Crique Alaparoubo, affluent du fleuve Sinnamary, a 3 km, 12 May 1969, de Granville J.J. 204 (MO!, P!); Riviere Petit Ouangi, 19 May 1973, de Granville J.J. 1864 (MO!, P!). FRENCH GUIANA. Saul, 11 Sep 1986, 3°38′N, 53°12′W, Freiberg M. 129 (B!); Saul, La Fumee Mt. La Fume West Trail, Alt. 200 m., 3°37′N, 53°12′W, 8 May 1986, Mori S.A, and Pennington T.D. 17932 (MO!, NY!, P!), Site du Barrage de Petit Saut (recolte’ vivant AFCPO 01-1994), cultive’ au jardin botanique Cherbourg (CH601), Parc Emmanuel Liais, 9 Feb 1999, Pignal M.M. 315 (P!).

Notes: The species is very similar to S. sessilis. The two can be distinguished by their leaf sheaths and floral bract surfaces. These structures are glabrous in S. yauaperyensis and dark-hirsute in S. sessilis.

Sobralia lowii Rolfe (Fig. 1E)

Gard. Chron. 2: 378. 1890; Type (Baranow 2015: 51): Colombia. Low H. & Co. s.n. (Lectotype: K! 364491; Isolectotypes: K! 364492, K! 364493, K! 364494, K! 364495, K! 364496).

Plant 30–45 cm tall, erect. Stem forming keikis, covered green, nearly smooth except the greenish hue leaf sheaths. Leaves 12–15 cm long, 2–3 cm wide, narrowly lanceolate, acuminate, minutely hirsute on the lower side along the veins. Floral bracts 20-30 mm long, lanceolate, acute, hirsute. Flower single at a time, bright purple, gynostemium white. Dorsal sepal 50 mm long, 23 mm wide, oblanceolate, acute. Lateral sepals 50 mm long, 24 mm wide, lanceolate, oblique, acute, apiculate. Petals ca. 45 mm long, 25 mm wide, lanceolate, acute, slightly oblique, margins undulate in apical half. Labellum ca 45 mm long, 33 mm long, elliptical, obtuse, apical half margins crispate-undulate, base with a pair of white, fleshy, linear calli. Column ca 40 mm long, erect, apical stelidia narrow, strongly recurved, acute. The middle of the column with two raised keels in front. Column with two raised keels in the middle of its front.

Habitat and ecology: No data.

Distribution: Colombia.

Representative specimens: Colombia [Cauca] New Grenada. Low H. & Co. s.n. (K!).

Notes. By its small habit and raised keels in front of the central part of the column, the species reminds S. decora. However, S. lowii has larger sepals (50 mm vs 40 mm in S. decora). Besides, the two species can be distinguished by the structure of the leaf sheaths –they are covered by minute, black hairs in S. lowii while in S. decora they are green and nearly smooth. The black hairs may place S. lowii close to S. sessilis, however the latter species is distinctly larger.

Sobralia decora Bateman (Fig. 1F)

Orchid. Mexico & Guatemala: t. 26. 1841; Type: Guatemala. Skinner G. s.n. (Holotype: K-L!).

= Sobralia neglecta Schltr., Repert. Spec. Nov. Regni Veg. Beih. 19: 161. 1923; Type (Barringer 1986: 18): Costa Rica. Brenes A.M. 279 (Lectotype: AMES! 00090616; Isolectotype: CR 26282; F –photo, AMES! 00104312 - drawing).

Erect, cane-like plants 30–60 cm tall, often growing in clumps. Stem leafy above, often forming keikis on the old inflorescences, leaf sheaths nearly smooth, green. Leaves up to 15 cm long, 2–4 cm wide, lanceolate, acuminate, plicate. Inflorescence terminal, cone-like, with 1, rarely 2 flowers at a time. Flowers small sized as for the genus, sepals and petals white or pale pink, labellum pink inside with pale reddish brown throat and a white spot at the apex, gynostemium white. Dorsal sepal 35–40 mm long, 12–15 mm wide, lanceolate, acute, curved backwards. Lateral sepals 35–40 mm long, 12–15 mm wide, lanceolate, acute, curved backwards. Petals broader and shorter than sepals, curved only at the extremities. Labellum 35–40 mm long, ca. 20 mm wide, with a pair of fleshy, ridge-like calli at the base, cucullate, obovate, curled at the margin, which is also bent outwards. Column 15 mm long, almost concealed by the lip, apical stelidia, strongly falcate, not exceeding the column apex. The middle of the column with two raised keels in front.

Habitat and ecology: Usually terrestrial in deep shaded ravine in leaf mould, on rocks in oak-pine forests, occasionally in the crotch of trees. Flowering in Colombia in January, July, August, and December.

Distribution: Mexico, Belize, Guatemala, El Salvador, Honduras, Nicaragua, Costa Rica, Panama, Colombia, Brazil. Alt. from sea level up to 500 m.

Representative specimens: COLOMBIA. Cauca. Mpio de Guapi, Parque Nal. Natural Gorgona - Alto de los Micos, 9 Sep 1987, Lozano C., Rangel O. & Estudiantes Postgrado 5724 (COL!); Chocó. Vicinity of Jequedo, 42 km W of Quibdó-Istmina Road on Pan American Highway under construction, 10 km of río Pato, Alt. 220 m. 10 Jan 1979, Gentry A. & Renteria E.A. 23961 (COL). Valle del Cauca. El Queremal, 20 Jan 1980, Guarin O. 51 (COL!). Vaupés. Yapoboda, 10 Dec 1943, Allen P.H. 31556 (MO!); Serrania de Traira. 10 km al NW del raudal de La Libertad, 0°58′S, 69°45′W, Alt. 250 m. 24 Jul 1993, Cortes R. & Rodriguez J. 547 (COL!); The same loc. Cortes R.& Rodriguez J. 572 (COL!); The same loc. 26 Jul 1993, Cortes R. & Rodriguez J. 596 (COL!); Serrania de Taraira, 20 km al NW del Raudal de la Libertad. 0°53′S, 69°45′W, Alt. 250 m. 4 Aug 1993, Cortes R. & Rodriguez J. 772 (COL!). Sine loc. Linden J.J. 2&3 (W!). BRAZIL. Amazonas. Rio Negro near Ponto de Paagadodo, in the Parana de Anavilhanas, 2°45′S, 60°55′W, 9 Jun 1990, Mori S., Gracie C., Betros H., Hecht S., Hecht J. van Etten M. & Wright F. 21226 (K!)

Notes. The species can be recognized by short stems which not exceed 60 cm in length. Besides, the leaves and leaf sheaths of S. decora are nearly smooth, sometimes with greenish hue only—this is an important character allowing to distinguish S. decora from S. lowii which is also dwarf in size if compared with remaining species of the complex. However, S. lowii has small but distinct black hairs on leaf sheaths.

Supplemental Information

Material S1 List of herbarium specimens cited in the manuscript

Material S2 Herbarium labels data used for the analysis of the distribution, habitat and ecology of the studied species

The data were used for the analysis of the distribution, habitat and ecology of the discussed taxa.

We would like to express our gratitude to the curators of the herbaria mentioned in the material and methods sections. We thank Mauro Rosim for providing the photographs published in this work.

Additional Information and Declarations

Competing Interests

Author Contributions

Data Availability

The authors declare there are no competing interests.

Przemyslaw Baranow conceived and designed the experiments, performed the experiments, analyzed the data, prepared figures and/or tables, authored or reviewed drafts of the article, and approved the final draft.

Dariusz L. Szlachetko conceived and designed the experiments, performed the experiments, prepared figures and/or tables, authored or reviewed drafts of the article, and approved the final draft.

The following information was supplied regarding data availability:

The raw data are available in the Supplementary File.

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
