# Peer review of "Sobralia decora Bateman (Orchidaceae) and its relatives in South America"

_PeerJ, doi:10.7717/peerj.18078_

## Round 0.1 · original submission · Major Revisions

Both reviewers raised many points of criticism. Please, fix these issues.

Reviewer 1 ·

Basic reporting

The English language used throughout should be improved. It is clear but often poor. Furthermore, the writing is often allusive to taxonomic contentions that are perhaps familiar to a very specialized public but otherwise completely unknown to the general reader.

Considering that the manuscript would be a taxonomic “clarification” of the group, the illustration apparatus is amply insufficient to show the differences between the species, and reduced to floral diagrams, in one case also incomplete (just two parts of the petals’ row). It would be advisable that at least a photograph of a living flower should be included in the manuscript, not to dream about multiple photographs to suggest the range of natural variation in the concerned species.

Raw data are supplied.

Experimental design

The manuscript surely represents original research, and should fill a knowledge gap, but the methodology is questionable. Taxa comparison is essentially based on the lecture of protologues and the study of original materials, plus the examination of 64 herbarium specimens, of which almost 50% belong (allegedly) to a single species, Sobralia sessilis. Due to their thin, aqueous, ephemeral tissues, the flowers of Sobralia are particularly well-known for making poor and sometimes useless specimens for herbaria. Fine details of the reproductive organs of Sobralia, which are used as key characters in the study, are virtually non discernible in pressed flowers. For this genus in particular the study of some living specimens should be considered a pre-requisite for research. From this point of view, I would consider that the technical standards adopted are poor.

Validity of the findings

The conclusions of the study are not unreasonable, but likely they are very incomplete and premature.

Additional comments

The group of species, as circumscribed here, is in my opinion both polyphyletic and paraphyletic, as it obviously includes in the group one of the species, S. biflora. that does not belong here, and probably leaves out at least another species that should be considered.
The adopted methodology is questionable, as the entire study did not include the examen of any single living specimen of the concerned group. This is particularly critical in the case of Sobralia, as the highly ephemeral tissues of its flowers make this genus a poor candidate for herbarium studies.

Annotated reviews are not available for download in order to protect the identity of reviewers who chose to remain anonymous.

Reviewer 2 ·

Basic reporting

This ms tries to clarify the inconsistencies for the taxonomic recognition of Sobralia (Orchidaceae) species in South America. The aim focus on the comparative morphological analysis obtained from the review of botanical vouchers from different herbaria located in several South American countries, leading to proposing an identification key for the different species. Far from this, it seems to me that there is no proposal or justification in the introduction of the relevance of carrying out this study. I suggest that more arguments be given as to why it is essential to have this new proposal for recognition between previously described species. I see the convenience of having a single document that summarizes the information on all included species, but I think it would be very interesting if the authors provided other arguments in favor of the study.

Experimental design

The study lacks an experimental design as it is a comparative morphological study. The theoretical and methodological foundations need to be further detailed.

Validity of the findings

Certainly, the similarity among any Sobralia species may be the reason for some misunderstandings of the species concepts and misidentification of some of the herbarium collections. The authors' proposal to name the Sobralia decora complex goes beyond South America, since it is well known that Sobralia decora and several related species are present in Central America and southeastern Mexico, so the treatment of the complex should also include to these other regions. It is advisable that the authors justify with more solid bases the reasons for having restricted the study only to South America. In my opinion, it is not enough to indicate that Dressler's (2012) study already addressed the variability of this species in Central America and then not relate the results to this document.

Additional comments

On the other hand, and considering recent integrative taxonomy studies, it seems to me that the morphological study approach alone does not allow us to identify possible phylogenetic relationships between species. Although this is not an objective per se, I suggest that the authors describe in the discussion the limitations of the study for the comprehensive understanding of the morphological, genetic, and functional relationships of Sobralia spp.

---

## Round 0.2 · accepted · Accept

I have carefully checked rebuttal letter and file with track changes. Authors replied satisfactorily to both reviewers